# ANGULAR VISUAL HARDNESS

## ABSTRACT

Although convolutional neural networks (CNNs) are inspired by the mechanisms behind human visual systems, they diverge on many measures such as ambiguity or hardness. In this paper, we make a surprising discovery: there exists a (nearly) universal score function for CNNs whose correlation is statistically significant than the widely used model confidence with human visual hardness. We term this function as angular visual hardness (AVH) which is given by the normalized angular distance between a feature embedding and the classifier weights of the corresponding target category in a CNN. We conduct an in-depth scientific study. We observe that CNN models with the highest accuracy also have the best AVH scores. This agrees with an earlier finding that state-of-art models tend to improve on classification of harder training examples. We find that AVH displays interesting dynamics during training: it quickly reaches a plateau even though the training loss keeps improving. This suggests the need for designing better loss functions that can target harder examples more effectively. Finally, we empirically show significant improvement in performance by using AVH as a measure of hardness in self-training methods for domain adaptation.

## 1 INTRODUCTION

The invention and development of Convolutional Neural Networks (CNNs) were inspired by natural visual processing systems. For example, artificial neurons were designed to mimic neurons taking and transforming information [48], and neocognitron, the origin of the CNN architecture, was enlightened by early findings of receptive fields in the visual cortex [15]. CNNs have achieved a great success in pushing the boundaries in a wide range of computer vision tasks such as image classification [24, 30, 59], face recognition [39, 63, 64], and scene analysis [19, 43, 70]. Specifically, on certain large-scale benchmarks such as ImageNet [10], CNNs have even surpassed human-level accuracy. Despite such notable progress, CNNs are still far from matching human vision on many measures such as robustness, adaptability and few-shot learning [23], and could suffer from various biases. For example, CNNs pre-trained on Imagenet are biased towards textures [18]. These biases can result in CNNs being overconfident, or prone to domain gaps and adversarial attacks. Therefore, to fundamentally solve the above problems, the efforts should be made to improve CNN's capabilities to model human visual system [47, 62].

The popular measure of CNN confidence, softmax score, has been widely used in many applications, yet causing calibration problems and tending to make CNNs overconfident even if they are wrong [21, 34]. However, this is not the case with human vision. Thus, there is a gap between the current measure of hard examples that appear to be ambiguous or uncertain in these two systems. We denote human visual hardness as the measure of how hard an example is to human visual system. In this paper, we bridge the gap by proposing a novel score function on CNNs that correlates closely with human visual hardness. The first piece of this puzzle starts with the question of what is a good measure of human visual hardness. Recently, [52] argued that human selection frequency is a good measure. This is the average number of times an image gets picked by a crowd of annotators, when they are asked to pick an image from a pool that belongs to a certain specified category. Intuitively, human selection frequency depends on various factors like object sizes, poses, special filters applied to images, etc. [52] collected human selection frequency scores on ImageNet validation set using the MTurk platform. In this paper, we use this dataset to verify several hypotheses on correlations between CNNs and human visual systems in section 3.2.

Moreover, an automatic detection of examples that are hard for human vision has numerous applications. [52] showed that state-of-the-art models perform better on hard examples (*i.e.*, hard for humans). This implies that in order to improve generalization, the models need to improve accuracy on hard examples. This can be achieved through various learning algorithms such as curriculum learning [2] and self-paced learning [32] where being able to detect hard examples is crucial. Measuring sample confidence is also important in partially-supervised problems such as semi-supervised learning [71, 72], unsupervised domain adaptation [8] and weakly-supervised learning [65] due to their under-constrained nature. For instance, self-training [73] can easily reach to trivial solutions if one does not select pseudo-labels carefully based on correct measure of hardness. Furthermore, by identifying hard examples, one can detect various biases in current CNN models. Sample hardness can also be used to identify implicit distribution imbalance in datasets to ensure fairness and remove societal biases [4].

**Our contributions are summarized as follows:**

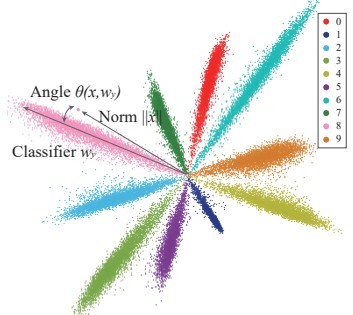

**Angular visual hardness (AVH):** Given a CNN, we propose a novel score function that has stronger correlation with human visual hardness than softmax score. It is the normalized angular distance between the image feature embedding and the weights of the target category (See Figure 1). The normalization takes into account the angular distances to other categories. We argue that the semantic ambiguity that affects human visual hardness has stronger correlation with this score and we find empirical evidence to support this claim.

Figure 1: Visualization of 2D CNN features which are output directly from the CNN by setting the feature dimension as 2 on MNIST.

The AVH score is inspired by the observation from Figure 1 and also [42] that samples from each class concentrate in a convex cone in the embedding space. In addition, some existing theoretical results [61] show that for minimization of logistic loss or cross-entropy loss which is often used in CNNs, gradient descent converges to the same direction as maximum margin solutions irrelevant to the $\ell_2$ norm of classifier weights or feature embeddings. This also provides the intuition behind why we can show AVH score rather than current model confidence, softmax score, correlates better with human visual hardness.

We validate that there is a statistically-significant stronger correlation between AVH and human selection frequency across a wide range of CNN models. Hence, it serves as its proxy on datasets where such information is not available and is beneficial to a number of downstream tasks.

We observed the evolution of AVH score during training of CNN models. It plateaus early in training even if the training (cross-entropy) loss keeps improving. This suggests the need to design better loss functions that can improve performance on hard examples. We also validate the argument in [52] that improving on hard examples is the key to improve the generalization by verifying that the state-of-the-art models have the best average AVH scores.

Finally, we empirically show the superiority of AVH with its application to self-training for unsupervised domain adaptation. With AVH being an improved confidence measure, our proposed self-training framework renders considerably improved pseudo-label selection and category estimation, leading to state-of-the-art results with significant performance gain over baselines.

## 2 RELATED WORK

**Example hardness measures:** Recently, measuring sample hardness for deep learning models has been widely studied with loss value [58], relative Euclidean distance [56, 66] and gradient norm [28]. On the other hand, there is a rich history in cognitive and neuroscience communities to understand human visual perception [6, 7, 14, 45], where many of them focus on mechanisms used by the human brain to translate visual information to mental representations. These representations are subject to many correspondence differences and errors and thereby are not isomorphic to the real world [37]. They can be affected by the ambiguity of different semantics [27] such as occlusion, distortion, motion blur, and inherent similarity among objects. Due to the expensive human labeling process, such detailed semantic information is typically not present in large-scale image benchmarks used to train the CNN models.

**Angular distance in CNNs:** [69] uses the deep features to quantify the semantic difference between images, indicating that deep features contain the most crucial semantic information. It empirically shows that the angular distance between feature maps in deep neural networks is very consistent with the human in distinguishing the semantic difference. However, because of the different goal mentioned above, they have not studied or shown any strong correlation of human visual hardness and the angular distance on natural images. [40] proposes a hyperspherical neural network that constrains the parameters of neurons on a unit hypersphere and uses angular similarity to replace the inner product similarity. [42] decouples the inner product as the norm and the angle and argues that the norm corresponds to intra-class variation, and the angle corresponds to inter-class semantic difference. However, this work does not consider any human factors, while our goal is to bridge the gap between CNNs and human perception. [35, 41] propose well-performing regularizations based on angular diversity to improve the network generalization.

**Image degradation:** Because CNNs and humans achieve similar accuracy on a wide range of tasks on benchmark datasets, a number of works have investigated similarities and differences between CNNs and human vision [3, 5, 9, 11, 12, 29, 46, 51, 67]. Since human annotation data is hard to come by, researchers have proposed an alternative measure of visual hardness on images based on image degradation [37]. It involves adding noise or changing image properties such as contrast, blurriness, and brightness. [17] employed psychological studies to validate the degradation method as a way to measure human visual hardness. It should be noted that the artificial visual hardness introduced by degradation is a different concept from the natural visual hardness. The hardness based on degradation only reflects the hardness of a single original image with various of transformations, while natural visual hardness based on the ambiguity of human perception across a distribution of natural images. In this paper, we consider both as the surrogates of human visual hardness.

**Deep model calibration**. Confidence calibration is the problem of predicting probability estimates representative of the true correctness likelihood [21]. It is well-known that the deep neural networks are mis-calibrated and there has been a rich literature trying to solve this problem [21, 31]. However, this is a somewhat different issue because the confidence calibration is a problem introduced by two measurements of the model, which does not have any involvement of human visual hardness.

## 3   A DISCOVERY FROM SCIENTIFIC TESTING: ANGULAR VISUAL HARDNESS

### 3.1   NOTATIONS AND SETUP

In order to quantify Human Visual Hardness and Model Predictions for convenience purposes in experiments, we use corresponding surrogates which are formally defined as the following throughout the paper. We use the ImageNet [10] benchmark in all following experiments. Particularly, we take advantage of the Human Selection Frequency information for validation images provided by the recent paper [52]. Recall that such information can serve as one of the proxy for Human Visual Hardness. To test if our findings with Human Selection Frequency hold on another proxy, image degradation, we create an augmented validation set based on two image degradation methods, decreasing contrast and adding noise. We label them with corresponding degradation level (results shown in Appendix A.2 and A.5). Besides, in order to verify that the our experimental results hold consistently across models instead of a particular model, we use four popular ImageNet pre-trained models AlexNet [30], VGG19 [59], DenseNet121 [26], ResNet50 [24]. We select ResNet50 as the representative model for some experiments. Most importantly, we also provide experimental results on different datasets, MNIST and CIFAR10/100, in Appendix A.3 and A.4 to better support our proposal.

Denote $\mathbb{S}^n$ as the unit $n$-sphere, formally, $\mathbb{S}^n = \{\boldsymbol{x} \in \mathbb{R}^{n+1} | \|\boldsymbol{x}\|_2 = 1\}$. Below by $\mathcal{A}(\cdot, \cdot)$, we denote the angular distance between two points on $\mathbb{S}^n$, i.e., $\mathcal{A}(\boldsymbol{u}, \boldsymbol{v}) = \arccos(\frac{\langle \boldsymbol{u}, \boldsymbol{v} \rangle}{\|\boldsymbol{u}\|\|\boldsymbol{v}\|})$. Let $x$ be the feature embeddings input for the layer before the last one of the classifier of the pretrained CNN models, eg. FC2 for VGG19. Let $\mathcal{C}$ be the number of classes for a classification task. Denote $\mathcal{W} = \{w_i | 0 < i \leq \mathcal{C}\}$ as the set of weights for all $\mathcal{C}$ classes in the final layer of the classifier.

**Definition 1** (Angular Visual Hardness (AVH)). *AVH, for any $(\boldsymbol{x}, \boldsymbol{y})$, is defined as:*

$$AVH(\mathbf{x}) = \frac{\mathcal{A}(\mathbf{x}, \mathbf{w}_y)}{\sum_{i=1}^{C} \mathcal{A}(\mathbf{x}, \mathbf{w}_i)},$$

*which $\mathbf{w}_y$ represents the weights of the target class.*

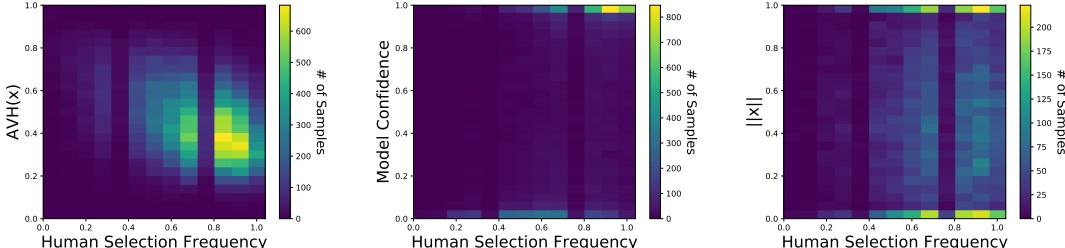

Figure 2: The left one presents Human Selection Frequency v.s. $AVH(\|x\|)$, which we can see strong correlation. The second plot presents the correlation between Human Selection Frequency and Model Confidence with ResNet50. It is not surprising that the density is highest on the right corner. The third one presents Human Selection Frequency v.s. $\|x\|$. There are no obvious correlation between them. Note that different color indicates the density of samples in that bin.

Table 1: presents the spearman rank correlations between human selection frequency and AVH, Model Confidence and L2 Norm of the Embedding in ResNet50 for different visual hardness bin of samples. Noted here we show the absolute value of the coefficient which represents the strength of the correlation. For example, $[0, 0.2]$ denotes the samples that have human selection frequency from 0 to 0.2.

| | z-score | Total Coef | $[0, 0.2]$ | $[0.2, 0.4]$ | $[0.4, 0.6]$ | $[0.6, 0.8]$ | $[0.8, 1.0]$ |
|---|---|---|---|---|---|---|---|
| Number of Samples | - | 29987 | 837 | 2732 | 6541 | 11066 | 8811 |
| AVH | 0.377 | 0.36 | 0.228 | 0.125 | 0.124 | 0.103 | 0.094 |
| Model Confidence | 0.337 | 0.325 | 0.192 | 0.122 | 0.102 | 0.078 | 0.056 |
| $\|x\|$ | - | 0.0017 | 0.0013 | 0.0007 | 0.0005 | 0.0004 | 0.0003 |

**Definition 2** (Model Confidence). *We define model confidence on a single sample as the probability score of the true objective class output by the CNN models, formally, $\frac{e^{\mathbf{w}_y \boldsymbol{x}}}{\sum_{i=1}^{C} e^{\mathbf{w}_i \boldsymbol{x}}}$.*

**Definition 3** (Human Selection Frequency). *We define one way to measure human visual hardness on pictures as Human Selection Frequency. Quantitatively, given $m$ number of human workers in a labeling process described in [52], if $b$ out of $m$ label a picture as a particular class and that class is the target class of that picture in the final dataset, then Human Selection Frequency is defined as $\frac{b}{m}$.*

## 3.2 CONNECTIONS AND GAPS BETWEEN HUMAN VISUAL SYSTEM AND CNN

Studying the precise connection or gap between human visual hardness and model predictions is not feasible because data collection involving human labelling or annotation requires large amount of work. In addition, usually those human data is application or dataset specific, which makes the scalability of this study even worse. Therefore, all the testing and experiments we design are at best effort given the limited resources. That is exactly another motivation for us to bridge the gap between Human and models because models predictions require minimum costs compared to human efforts. In this section, We first provide four hypothesis and test them accordingly.

**Hypothesis 1.** *AVH has a correlation with Human Selection Frequency.*

**Outcome: Null Hypothesis Rejected**

Correspondingly, after evaluating each validation sample on pre-trained models, we extract feature embeddings $x$ and also the class weights $\mathcal{W}$ to compute **AVH**$(x)$. Noted that we linear scale the range of **AVH**$(x)$ to $[0, 1]$. Table 1 shows the overall strong correlation of **AVH**$(x)$ and Human Selection Frequency consistently (p-value is $< 0.001$ rejecting the null hypothesis). From the coefficients represented for different bins of example hardness, we can see that the harder the examples, the weaker the correlation. Noted that we also check the results across four different CNN architectures and we found that better model has higher coefficient.

The plot on the left in Figure 2 help visualize the strong correlation between **AVH**$(x)$ and Human Selection Frequency for validation images. One intuition behind this correlation is that the class weights $W$ might corresponds to human semantic for each category and thereby **AVH**$(x)$ corresponds to human semantic categorization of an image. In order to test if the strong correlation holds for all models, we perform the same experiments on AlexNet, VGG19 and DenseNet121.

**Hypothesis 2.** *Model Confidence has a correlation with Human Selection Frequency.*

**Outcome: Null Hypothesis Rejected**

An interesting observation in [52] shows that Human Selection Frequency has strong influence on the Model Confidence. Specifically, examples with low Human Selection Frequency tends to have relatively low Model Confidence. Naturally we examine if the correlation between Model Confidence and Human Selection Frequency is strong. Specifically, all ImageNet validation images are evaluated by the pre-trained models. The corresponding output is simply the Model Confidence on each image. From table 1, we can first see that it is clear that because p-value is $< 0.001$, Model Confidence does have a strong correlation with Human Selection Frequency. However, the correlation coefficient for Model Confidence and Human Selection Frequency is consistently lower than that of AVH and Human Selection Frequency.

The middle plot in figure 2 presents a two-dimensional histogram for the correlation visualization. The x-axis represents Human Selection Frequency, and the y-axis represents Model Confidence. Each bin exhibits the number of images which lie in the corresponding range. We can observe the high density at the right corner, which means the majority of the images have both high human and model accuracy. However, there is a considerable amount of density on the range of medium human accuracy but either extremely low or high model accuracy. One may question that the difference of the correlation coefficient is not large, thereby we also run statistical testing on the significance of the gap, naturally our next step is to test if the difference is significant.

**Hypothesis 3.** *AVH has a stronger correlation to Human Selection Frequency than Model Confidence.*

**Outcome: Null Hypothesis Rejected**

There are three steps for testing if two correlation coefficient are significantly different. First step is applying Fisher Z-Transformation to both coefficient. The Fisher Z-Transformation is a way to transform the sampling distribution of the correlation coefficient so that it becomes normally distributed. Therefore, we apply fisher transformation for each correlation coefficient: $Z$ score for coefficient of AVH becomes $0.377$ and that of Model Confidence becomes $0.337$. The second step is to compute the $Z$ value of two $Z$ scores. Then we determined the $Z$ value to be $4.85$ from the two above-mentioned $Z$ scores and sample sizes. The last step is that find out the p-value according to the $Z$ table. According to $Z$ table, p-value is $0.00001$. Therefore, we reject the null hypothesis and conclude that AVH has statistically significant stronger correlation with Human Selection Frequency than Model Confidence. In later section 5, we also empirically show that such stronger correlation brings cumulative advantages in some applications. In appendix A.1, besides Spearman correlation coefficient, we have also shown Pearson and Kendall Tau ones. Further more, to test if the conclusion holds for different models, we run the same tests on all four different architectures. The conclusion is for all the models and under different testings, AVH correlate significantly stronger than model confidence, but the correlation is even stronger for better models.

**Hypothesis 4.** $\|x\|_2$ *has a correlation with Human Selection Frequency.*

**Outcome: Failure to Reject Null Hypothesis**

[42] conjectures that $\|x\|_2$ accounts for intra-class Human/Model Confidence. Particularly, if the norm is larger, the prediction from the model is also more confident, to some extent. Therefore, we conduct similar experiments like previous section to demonstrate the correlation between $\|x\|_2$ and Human Selection Frequency. Initially, we compute the $\|x\|_2$ for every validation sample for all models. Then we normalize $\|x\|_2$ within each class. Table 1 presents the results for the correlation test. We omit the results for p-value in the table and report here that they are all much higher than $0.05$, indicating there is no correlation between $\|x\|_2$ and Human Selection Frequency.

The right plot in figure 2 uses a two-dimensional histogram to show the correlation for all the validation images. Given that the norm has been normalized with each class, naturally, there is notable density when the norm is $0$ or $1$. Except for that, there is no obvious correlation between $\|x\|_2$ and Human Selection Frequency. We further verify if presenting all samples across 1000 different classes affects the visualization of the correlation. According to WordNet [13] hierarchy, we map the original 1000 fine-grained classes to 45 higher hierarchical classes. A figure in appendix exhibits the relationship between Human Selection Frequency and $\|x\|_2$ for three representative higher classes containing $58, 7, 1$ fine-grained classes respectively. Noted that there is still not any visible direct proportion between these two variables across all plots.

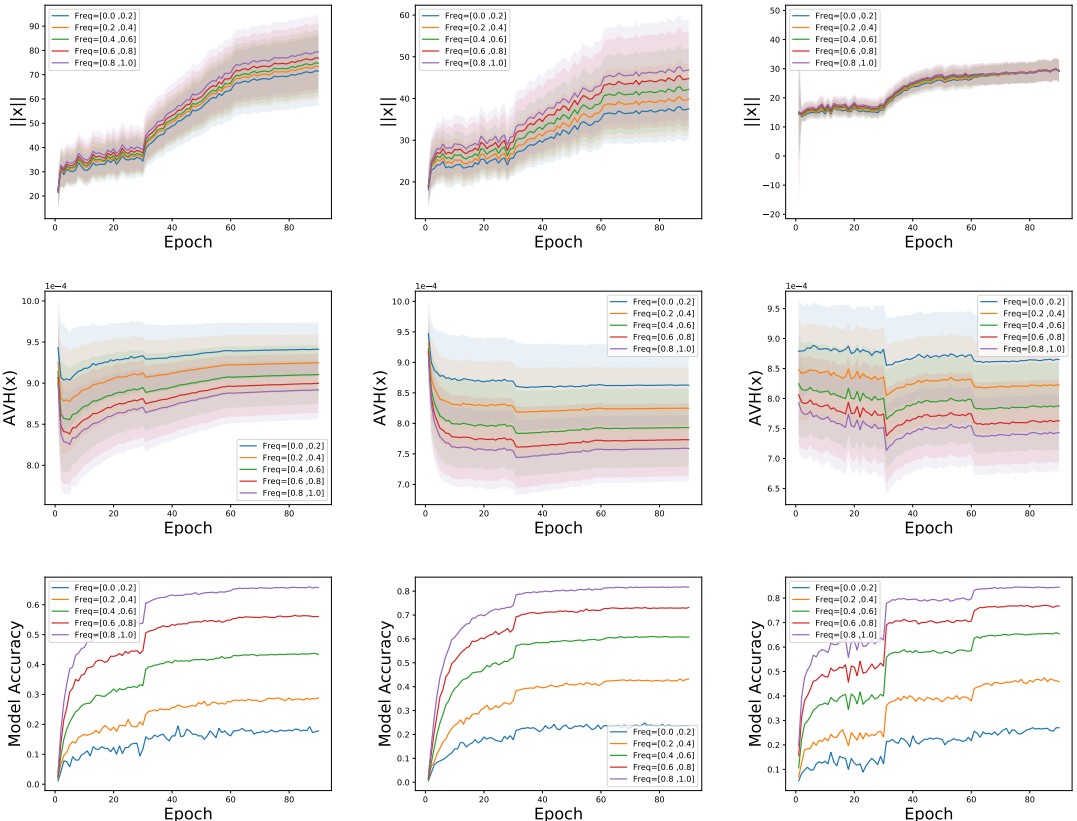

Figure 3: The top three plots show the number of Epochs v.s. Average $\ell_2$ norm across ImageNet validation samples which are split into five bins based on human selection frequency information. The middle three plots represent number of Epochs v.s. Average AVH(x). The bottom ones present number of Epochs v.s. Model Accuracy. From left to right, we use AlexNet, Vgg19 and ResNet50. The plots for DenseNet are in Appendix.

Besides the hypothesis testings, we provide a detailed discussion on the difference between AVH and Model Confidence in Appendix C.

## 4   DYNAMICS OF AVH DURING TRAINING

After discovering the strong correlation of human visual hardness and AVH score, a natural question would be: What role does AVH play during the training process? Optimization algorithms are used to update weights and biases i.e. the internal parameters of a model to improve the training loss. Both the angles between the feature embedding and classifiers, and the $L_2$ norm of the embedding can influence the loss. While it is well-known that the training loss or accuracy keeps improving but it is not obvious what would be the dynamics of the angles and norms separately during training. we design the experiments to observe the training dynamics of various network architectures.

**Experiment Settings**. For datasets and models, we use exactly the same setting as the experiments in 3.1. Nevertheless, observing training dynamics involves training models from scratch on ImageNet training set instead of directly using the pre-trained models. Therefore, we follow the standard training process of AlexNet [30], VGG19 [59], ResNet50 [24] and DenseNet121 [26] (DenseNet results are put in Appendix). For consistency, we train all four models for 90 epochs and decay the initial learning rate by a factor of 10 every 30 epochs. The initial learning rate for AlexNet and VGG19 is 0.01 and for DensetNet121 and ResNet50 is 0.1. For human visual hardness based on Human Selection Frequency, we split all the validation images into 5 bins, [0.0, 0.2], [0.2, 0.4], [0.4, 0.6], [0.6, 0.8], [0.8, 1.0], based on their human selection frequency respectively. For human visual hardness based on Image Degradation Level, we create an augmented validation set based on two image degradation methods,

decreasing contrast and adding noise. We label them with corresponding degradation level as well. Note that for all the figures in this section, Epoch starts from 1.

**Observation 1: The norm of feature embeddings keeps increasing during training**. Figure 3, 10 and 11 present the dynamics of the average $\|x\|_2$ and the dynamics of the accuracy for validation samples vary in 90 epochs during the training on three architectures. Note that we are using the validation data for dynamics observation and therefore have never fit them into the model. The average $\|x\|_2$ increases with a small initial slope but it suddenly climbs after 30 epochs when the first learning rate decay happens. The accuracy curve is very similar to that of the average $\|x\|_2$. The above observations are consistent in all models. More interestingly, we find that neural networks with shortcut connections (*e.g.*, ResNets and DenseNets) tend to make the norm of the images with different human selection frequency become the same, while the neural networks without shortcuts (*e.g.*, AlexNet and VGG) tend to keep the gap of norm among the images with different human visual hardness.

**Observation 2: AVH hits a plateau very early even when the accuracy or loss is still improving**. Figure 3, 10 and 11 exhibits the change of average AVH for validation samples in 90 epochs of training on three models. The average AVH for AlexNet and VGG19 decreases sharply at the beginning and then starts to bounce back a little bit before converging. However, the dynamics of the average AVH for DenseNet121 and ResNet50 are different. They both decrease slightly and then quickly hits a plateau in all three learning rate decay stages. But the common observation is that they all stop improving even when $\|x\|_2$ and model accuracy are increasing. AVH is more important than $\|x\|_2$ in the sense that it is the key factor deciding which class the input sample is classified to. However, optimizing the norm under the current softmax cross-entropy loss would be easier so, which cause the plateau of angles for easy examples. However, the plateau for the hard examples can be caused by the limitation of the model itself. As a result, it shows the necessity and importance of designing loss functions that focus on optimizing angles, such as [35, 39, 41].

**Observation 3: AVH's correlation with human selection frequency consistently holds across models throughout the training process**. In Figure 3, 10 and 11, we average over validation samples in five human selection frequency bins or five degradation level bins separately , and then compute the average embedding norm, AVH and model accuracies. We can observe that for $\|x\|_2$, the gaps between the samples with different human visual hardness are not obvious in ResNet and DenseNet, while they are quite obvious in AlexNet and VGG. However, for AVH, such AVH gaps are very significant and consistent across every network architecture during the entire training process. Interestingly, even if the network is far from being converged, such AVH gaps are still consistent across different human selection frequency. Also the norm gaps are also consistent. The intuition behind this could be that the angles for hard examples are much harder to decrease and probably never in the region for correct classification. Therefore the corresponding norms would not increase otherwise hurting the loss. It validates that AVH is a consistent and robust measure for visual hardness (and even generalization).

**Observation 4: AVH is an indicator of model's generalization ability**. From Figure 3, 18, 10 and 11, we observe that better models (*i.e.*, higher accuracy) have lower average AVH throughout the training process and also across samples under different human visual hardness. For instance, Alexnet is the worst model, and its overall average AVH and average AVH on each of five bins are worse than those of the other three models. In addition, we have found when testing Hypothesis 3 for better models, its AVH correlation is much more stronger than its Model confidence correlation with Human Selection Frequency. The above observations are aligned with the earlier observations of [52] that better models also generalize better on samples across different human visual hardness. Moreover, we AVH is potentially a better measure for generalization as a pretrained model. The norm of feature embeddings is often embedded with training data prior such as data imbalance [39] and class granularity [30]. But when extracting the features for the classes that do not exist in training set, such training data prior is undesired. Since AVH does not consider the norm of feature embeddings, it may better evaluate the generalization of the deep network.

**Conjecture on training dynamics of CNNs**. From Figure 3 and observations above, we conjecture that the training of CNN has two phases. 1) At the beginning of the training, the softmax cross-entropy loss will first optimize the angles among different classes while the norm will fluctuate and increase very slowly. We argue that it is because changing the norm will not decrease the loss when the angles are not separated enough for correct classification. As a result, the angles get optimized firstly. 2)

As the training continues, the angles become more stable and change very slowly while the norm increases rapidly. On the one hand, for easy examples, it is because when the angles get decreased enough for correct classification, the softmax cross-entropy loss can be well minimized by purely increasing the norm. On the other hand, for hard examples, the plateau is caused by that the CNN is unable to decrease the angle to correctly classify examples and thereby also unable to increase the norms (because it may otherwise increase the loss).

## 5 APPLICATION TO SELF-TRAINING FOR DOMAIN ADAPTATION

Unsupervised domain adaptation [1] presents an important transfer learning problem with wide applications. Deep self-training [33] recently emerged as a powerful framework towards addressing this problem [53, 57, 73, 74]. Here we show the application of AVH as an improved confidence measure in self-training that could significantly benefit the domain adaptation task.

**Dataset:** We conduct expeiments on the VisDA-17 [50] dataset which is a widely used major benchmark for domain adaptation in image classification. The dataset contains a total number of $152,409$ 2D synthetic images from 12 categories in the source training set, and $55,400$ real images from MS-COCO [36] with the same set of categories as the target domain validation set. We follow the protocol of previous works to train a source model with the synthetic training set, and report the model performance on target validation set upon adaptation.

**Baseline:** We choose class-balanced self-training (CBST) [73] as our starting self-training baseline considering its good performance on domain adaptaiton. We also compare our model with confidence regularized self-training (CRST)[1] [74], a more recent state-of-the-art self-training framework improved over CBST with network prediction/pseudo-label regularized with smoothness. Specifically, our work follows the exact implementation of CBST/CRST in [74].

Specifically, given the labeled source domain training set $\mathbf{x}_s \in \mathbf{X}_S$ and the unlabeled target domain data $\mathbf{x}_t \in \mathbf{X}_T$, with known source labels $\mathbf{y}_s = (y_s^{(1)}, ..., y_s^{(K)}) \in \mathbf{Y}_S$ and unknown target labels $\hat{\mathbf{y}}_t = (\hat{y}_t^{(1)}, ..., \hat{y}_t^{(K)}) \in \hat{\mathbf{Y}}_T$ from $K$ classes, CBST performs joint network learning and pseudo-label estimation by treating pseudo-labels as discrete learnable latent variables with the following loss:

$$\min_{\mathbf{w}, \hat{\mathbf{Y}}_T} \mathcal{L}_{CB}(\mathbf{w}, \hat{\mathbf{Y}}) = -\sum_{s \in S} \sum_{k=1}^{K} y_s^{(k)} \log p(k|\mathbf{x}_s; \mathbf{w}) - \sum_{t \in T} \sum_{k=1}^{K} \hat{y}_t^{(k)} \log \frac{p(k|\mathbf{x}_t; \mathbf{w})}{\lambda_k} \quad (1)$$
$$\text{s.t. } \hat{\mathbf{y}}_t \in \mathbf{E}^K \cup \{\mathbf{0}\}, \forall t$$

where the feasible set of pseudo-labels is the union of $\{\mathbf{0}\}$ and the $K$ dimensional one-hot vector space $\mathbf{E}^K$, and $\mathbf{w}$ and $p(k|\mathbf{x}; \mathbf{w})$ represent the network weights and the classifier's softmax probability for class $k$, respectively. In addition, $\lambda_k$ serves as a class-balancing parameter controlling the pseudo-label selection of class $k$, and is determined by the softmax confidence ranked at portion $p$ (in descending order) among samples predicted to class $k$. Therefore, only one parameter $p$ is used to determine all $\lambda_k$'s. The optimization problem in (1) can be solved via minimizing with respect to $\mathbf{w}$ and $\hat{\mathbf{Y}}$ alternatively, and the solver of $\hat{\mathbf{Y}}$ can be written as:

$$\hat{y}_t^{(k)*} = \begin{cases} 1, \text{if } k = \arg\max_k \{\frac{p(k|\mathbf{x}_t; \mathbf{w})}{\lambda_k}\} \text{ and } p(k|\mathbf{x}_t; \mathbf{w}) > \lambda_k \\ 0, \text{otherwise} \end{cases} \quad (2)$$

The optimization with respect to $\mathbf{w}$ is simply normal network re-training with source labels and estimated pseudo-labels. And the complete self-training process involves alternatively repeat of network re-training and pseudo-label estimation.

**CBST+AVH:** We seek to improve the pseudo-label solver with better confidence measure from AVH. To this end, we propose the following definition of angular visual confidence (AVC) to represent the predicted probability of class $c$:

$$AVC(c|\mathbf{x}; \mathbf{w}) = \frac{\pi - \mathcal{A}(\mathbf{x}, \mathbf{w}_c)}{\sum_{k=1}^{K}(\pi - \mathcal{A}(\mathbf{x}, \mathbf{w}_k))}, \quad (3)$$

---

[1] We consider MRKLD+LRENT which is reported to be the highest one in [74].

Table 2: Statistics of the examples selected by CBST+AVH and CBST/CRST.

|  | TP Rate | AVH (avg) | Model Confidence(avg) | Norm $\|x\|$ (avg) |
|---|---|---|---|---|
| CBST+AVH | 0.844 | 0.118 | 0.961 | 20.84 |
| CBST/CRST | 0.848 | 0.117 | 0.976 | 21.28 |

and the pseudo-label estimation in CBST+AVH is accordingly defined as:

$$\hat{y}_t^{(k)*} = \begin{cases} 1, \text{ if } k = \arg\max_k\{\dfrac{AVC(k|\mathbf{x}_t;\mathbf{w})}{\lambda_k}\} \text{ and } AVC(k|\mathbf{x}_t;\mathbf{w}) > \lambda_k \\ 0, \text{ otherwise} \end{cases} \tag{4}$$

where $\lambda_k$ is differently determined by referring to $AVC(k|\mathbf{x}_t;\mathbf{w})$ ranked at portion $p$ among samples predicted to class $k$ by AVH, following a similar definition of $\lambda_k$ in CBST. In addition, network re-training in CBST+AVH follows the softmax self-training loss in (1).

One could see that AVH changes the self-training behavior from two ways with the conditions in (4):
**Improved selection:** This is determined by $AVC(k|\mathbf{x}_t;\mathbf{w}) > \lambda_k$.
**Improved classification:** This is determined by $k = \arg\max_k\{\frac{AVC(k|\mathbf{x}_t;\mathbf{w})}{\lambda_k}\}$.
Specifically, the former determines which samples are not ignored during self-training based on AVC, whereas the latter determines the pseudo-label class by taking argmax over normalized AVC scores. With calibrated confidence that better resembles human visual hardness, both aspects are likely to considerably influence the performance of self-training.

**Experimental Results:** We present the results of the proposed method in Table 3, and also show its performance with respect to different self-training epochs in Figure 4. One could see that CBST+AVH outperforms both CBST and CRST by a very significant margin. We would like to emphasize that this is a very compelling result under "apples to apples" comparison with the same source model, implementation and hyper-parameters.

**Analysis:** A major challenge of self-training is the amplification of error due to misclassified pseudo-labels. Therefore, traditional self-training methods such as CBST often use model confidence as the confidence measure to select confidently labeled examples. The hope is that higher confidence potentially implies lower error rate. While this in generally proves useful, the model tends to focus on the "less informative" samples, whereas ignoring the "more informative", harder ones near classier boundaries that could be essential for learning a stronger classifier.

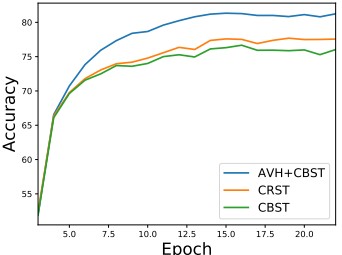

Figure 4: Experimental results of comparison among AVH+CBST, CBST and CRST on VisDA2017 dataset.

An advantage we observe from AVH is that the improved calibration leads to more frequent sampling of harder samples, whereas the pseudo-label classification on these hard samples generally outperforms softmax results. Table 2 shows some statistics of the examples selected with AVH and Model Confidence respectively at the beginning of the training process. The true postive rate (TP Rate) for of CBST+AVH remains similar to CBST/CRST, indicating AVH is not overall introducing more noisy examples compare to model confidence. On the other hand, it is observed that the average model confidence of AVH selected samples is lower, indicating there are more selected hard samples that are closer to the decision boundary. It is also observed that the average sample norm by AVH is also lower, confirming the influence of sample norm on ultimate model confidence.

## 6 EXTENSIONS AND APPLICATIONS

**Adversarial Example: A Counter Example?** Our claim about the strong correlation between AVH score and human visual hardness does not apply on non-natural images such as adversarial examples. For such examples, the human can not tell the difference visually, but the adversarial example has a worse AVH than the original image, which runs counter to our claim that AVH has strong correlation with human visual hardness. So this claim is limited to distribution of natural images. However, on a positive note, we do find that AVH is slower to change compared to the embedding norm during the dynamics of adversarial training. See Appendix for details.

Table 3: Experimental results on VisDA17.

| Method | Aero | Bike | Bus | Car | Horse | Knife | Motor | Person | Plant | Skateboard | Train | Truck | Mean |
|--------|------|------|-----|-----|-------|-------|-------|--------|-------|------------|-------|-------|------|
| Source [55] | 55.1 | 53.3 | 61.9 | 59.1 | 80.6 | 17.9 | 79.7 | 31.2 | 81.0 | 26.5 | 73.5 | 8.5 | 52.4 |
| MMD [44] | 87.1 | 63.0 | 76.5 | 42.0 | 90.3 | 42.9 | 85.9 | 53.1 | 49.7 | 36.3 | **85.8** | 20.7 | 61.1 |
| DANN [16] | 81.9 | 77.7 | 82.8 | 44.3 | 81.2 | 29.5 | 65.1 | 28.6 | 51.9 | 54.6 | 82.8 | 7.8 | 57.4 |
| ENT [20] | 80.3 | 75.5 | 75.8 | 48.3 | 77.9 | 27.3 | 69.7 | 40.2 | 46.5 | 46.6 | 79.3 | 16.0 | 57.0 |
| MCD [54] | 87.0 | 60.9 | **83.7** | 64.0 | 88.9 | 79.6 | 84.7 | 76.9 | 88.6 | 40.3 | 83.0 | 25.8 | 71.9 |
| ADR [55] | 87.8 | 79.5 | **83.7** | 65.3 | **92.3** | 61.8 | **88.9** | 73.2 | 87.8 | 60.0 | 85.5 | 32.3 | 74.8 |
| Source [74] | 68.7 | 36.7 | 61.3 | **70.4** | 67.9 | 5.9 | 82.6 | 25.5 | 75.6 | 29.4 | 83.8 | 10.9 | 51.6 |
| CBST [74] | 87.2 | 78.8 | 56.5 | 55.4 | 85.1 | 79.2 | 83.8 | 77.7 | 82.8 | **88.8** | 69.0 | **72.0** | 76.4 |
| CRST [74] | 88.0 | 79.2 | 61.0 | 60.0 | 87.5 | 81.4 | 86.3 | 78.8 | 85.6 | 86.6 | 73.9 | 68.8 | 78.1 |
| Proposed | **93.3** | **80.2** | 78.9 | 60.9 | 88.4 | **89.7** | 88.9 | **79.6** | **89.5** | 86.8 | 81.5 | 60.0 | **81.5** |

**Connection to deep metric learning:** Measuring the hardness of samples is also of great importance in the field of deep metric learning [49, 60, 66]. For instance, objective functions in deep metric learning consist of *e.g.*, triplet loss [56] or contrastive loss [22], which requires data pair/triplet mining in order to perform well in practice. One of the most widely used data sampling strategies is semi-hard negative sample mining [56] and hard negative sample mining. These negative sample mining techniques highly depend on how one defines the hardness of samples. AVH can be potentially useful in this setting.

**Connections to fairness in machine learning:** Easy and hard samples can implicitly reflect imbalances in latent attributes in the dataset. For example, the CASIA-WebFace dataset [68] mostly contains white celebrities, so the neural network trained on CASIA-WebFace is highly biased against the other races. [4] demonstrates a performance drop of faces of darker people due to the biases in the training dataset. In order to ensure fairness and remove dataset biases, the ability to identify hard samples automatically can be very useful. We would like to test if AVH is effective in these settings.

**Connections to knowledge transfer and curriculum learning:** The efficiency of knowledge transfer [25] is partially determined by the sequence of input training data. [38] theoretically shows feeding easy samples first and hard samples later (known as curriculum learning) can improve the convergence of model. [2] also show that the curriculum of feeding training samples matters in terms of both accuracy and convergence. We plan to investigate the use of AVH metric in such settings.

## 7 CONCLUDING REMARKS

Human perception and deep neural networks in general have different notions of visual hardness. Our paper studies the gap between them, and attempts to bridge this gap by proposing a novel measure for CNN models known as angular visual hardness. Our comprehensive empirical studies show that AVH has many nice properties. First, AVH has a strong correlation with human selection frequency and image degradation level. Second, this holds across different network architectures and throughout the training process. Third, AVH can serve as an indicator of generalization abilities of neural networks, and improving SOTA accuracy entails improving accuracy on hard examples. Then we empirically show the huge advantage of AVH over Model Confidence in self-training for domain adaptation task. It is still an open problem of designing an appropriate loss function that can focus on improving AVH during training. AVH can be very useful in other applications such as deep metric learning, fairness, knowledge transfer, etc. and we plan to investigate them in future.

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

# Appendix

## A    ADDITIONAL EXPERIMENTS

### A.1    ADDITIONAL RESULTS FOR CORRELATION TESTINGS

In order to run rigorous correlation testings, besides computing the Spearman coefficient, we provide additional results on Pearson and Kendall Tau correlation coefficients. Moreover, we show results for all four architectures, AlexNet, VGG19, ResNet50 and DenseNet121 in Table 4, 5, 6 and 7 respectively to support our claims in section 3.2.

### A.2    ADDITIONAL RESULTS FOR THE HYPOTHESIS

**Definition 4** (Image Degradation Level). *We define another way to measure human visual hardness on pictures as Image Degradation Level. We consider two degradation methods in this paper, decreasing contrast and adding noise. Quantitatively, Image Degradation Level for decreasing contrast is directly the contrast level. Image Degradation Level for adding noise is the amount of pixel-wise additive uniform noise.*

#### Hypothesis: AVH has a strong correlation with Image Degradation Level

In order to test if the results from Prediction 1 hold on another proxy to human visual hardness, Image Degradation Level, we perform the similar experiments but on the augmented ImageNet validation set. The plots in Figure 5 show the strong correlation between $\mathbf{AVH}(x)$ and Noise Degradation Level while the plots in Figure 6 present the strong correlation between $\mathbf{AVH}(x)$ and Contrast Degradation Level. They, along with Figure 7, demonstrate that $\mathbf{AVH}(x)$ strongly correlates with Human Visual Hardness. Additional Plots for DenseNet121 is shown in Figure 8.

#### Hypothesis: $\|x\|_2$ has a correlation with Human Selection Frequency

We further verify if presenting all samples across 1000 different classes affects the visualization of the correlation. According to WordNet [13] hierachy, we map the original 1000 fine-grained classes to 45 higher hierarchical classes. Figure 9 exhibits the relationship between Human Selection Frequency and $\|x\|_2$ for three representative higher classes containing $58, 7, 1$ fine-grained classes respectively. Noted that there is still not any visible direct proportion between these two variables across all plots.

#### Hypothesis: AVH has a correlation with Human Selection Frequency Additional results for DenseNet121 are shown in Figure 8.

Table 4: presents the spearman rank correlations between human selection frequency and AVH, Model Confidence in AlexNet. Noted here we show the absolute value of the coefficient which represents the strength of the correlation. Z value is computed by Z scores of both coefficients. p-value< 0.05 represents the result is significant.

|  | Coef with AVH | Coef with Model Confidence | $Z_{avh}$ | $Z_{mc}$ | Z value | p-value |
|---|---|---|---|---|---|---|
| Spearman | 0.339 | 0.325 | 0.352 | 0.337 | 1.92 | 0.027 |
| Pearson | 0.324 | 0.31 | 0.336 | 0.320 | 1.90 | 0.028 |
| Kendall Tau | 0.244 | 0.23 | 0.249 | 0.234 | 1.81 | 0.035 |

Table 5: presents the spearman rank correlations between human selection frequency and AVH, Model Confidence in VGG19. Noted here we show the absolute value of the coefficient which represents the strength of the correlation. Z value is computed by Z scores of both coefficients. p-value< 0.05 represents the result is significant.

|  | Coef with AVH | Coef with Model Confidence | $Z_{avh}$ | $Z_{mc}$ | Z value | p-value |
|---|---|---|---|---|---|---|
| Spearman | 0.349 | 0.335 | 0.364 | 0.348 | 1.94 | 0.026 |
| Pearson | 0.358 | 0.343 | 0.374 | 0.357 | 2.09 | 0.018 |
| Kendall Tau | 0.244 | 0.229 | 0.249 | , 0.233 | 1.94 | 0.026 |

Table 6: presents the spearman rank correlations between human selection frequency and AVH, Model Confidence in ResNet50. Noted here we show the absolute value of the coefficient which represents the strength of the correlation. Z value is computed by Z scores of both coefficients. p-value< 0.05 represents the result is significant.

| | Coef with AVH | Coef with Model Confidence | $Z_{avh}$ | $Z_{mc}$ | Z value | p-value |
|---|---|---|---|---|---|---|
| Spearman | 0.360 | 0.325 | 0.377 | 0.337 | 4.85 | < .00001 |
| Pearson | 0.385 | 0.341 | 0.406 | 0.355 | 6.2 | < .00001 |
| Kendall Tau | 0.257 | 0.231 | 0.263 | 0.235 | 3.38 | .0003 |

Table 7: presents the spearman rank correlations between human selection frequency and AVH, Model Confidence in DenseNet121. Noted here we show the absolute value of the coefficient which represents the strength of the correlation. Z value is computed by Z scores of both coefficients. p-value < 0.05 represents the result is significant.

| | Coef with AVH | Coef with Model Confidence | $Z_{avh}$ | $Z_{mc}$ | Z value | p-value |
|---|---|---|---|---|---|---|
| Spearman | 0.367 | 0.329 | 0.4059 | 0.355 | 6.2 | < .00001 |
| Pearson | 0.390 | 0.347 | 0.412 | 0.362 | 6.09 | < .00001 |
| Kendall Tau | 0.262 | 0.234 | 0.268 | 0.238 | 3.65 | .0001 |

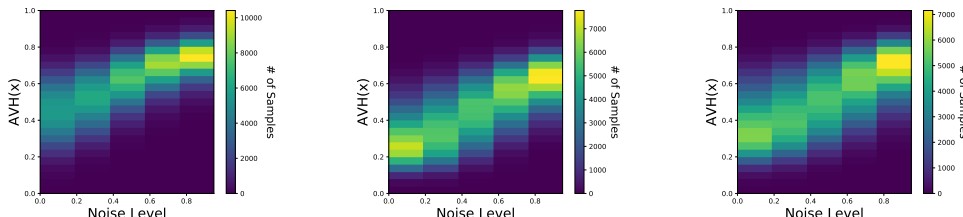

Figure 5: The three plots present the correlation between Noise Degradation Level and AVH using AlexNet, VGG19 and ResNet50 (DenseNet121 in Appendix). Noted here, the larger the Noise Level is, the harder human can visualize the image.



Figure 6: The three plots present the correlation between Contrast Degradation Level and AVH using AlexNet, VGG19 and ResNet50 (DenseNet121 in Appendix). Noted here, the larger the Contrast Level is, the easier human can visualize the image.

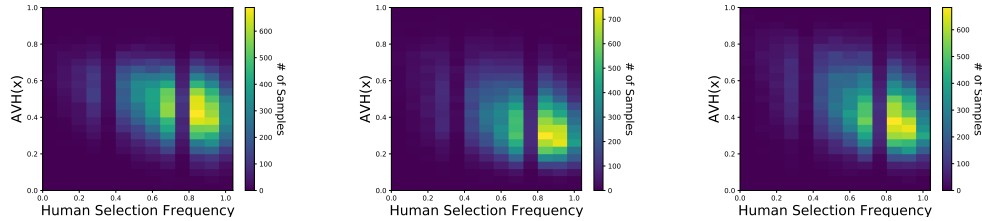

Figure 7: The three plots present the correlation between Human Selection Frequency and AVH using AlexNet, VGG19 and ResNet50 (DenseNet121 in Appendix).

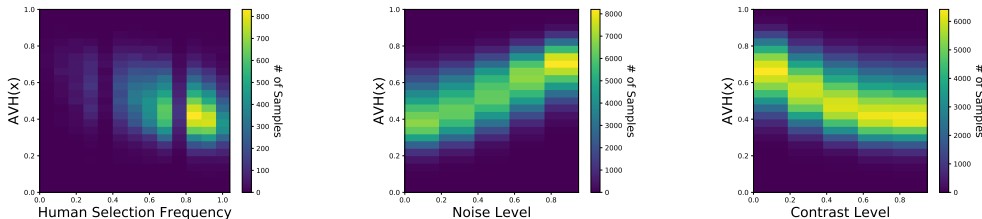

Figure 8: The left, middle and right plots respectively present the correlation between Human Selection frequency, Noise Degradation Level, Contrast Degradation Level and $\|x\|$ using DenseNet121.



Figure 9: $\ell_2$ norm of the embedding v.s. human selection frequency under different class granularity (according to WordNet hierarchy). From left to right, there are 58, 7, 1 classes respectively. The human selection frequency is therefore computed based on the new class granularity.

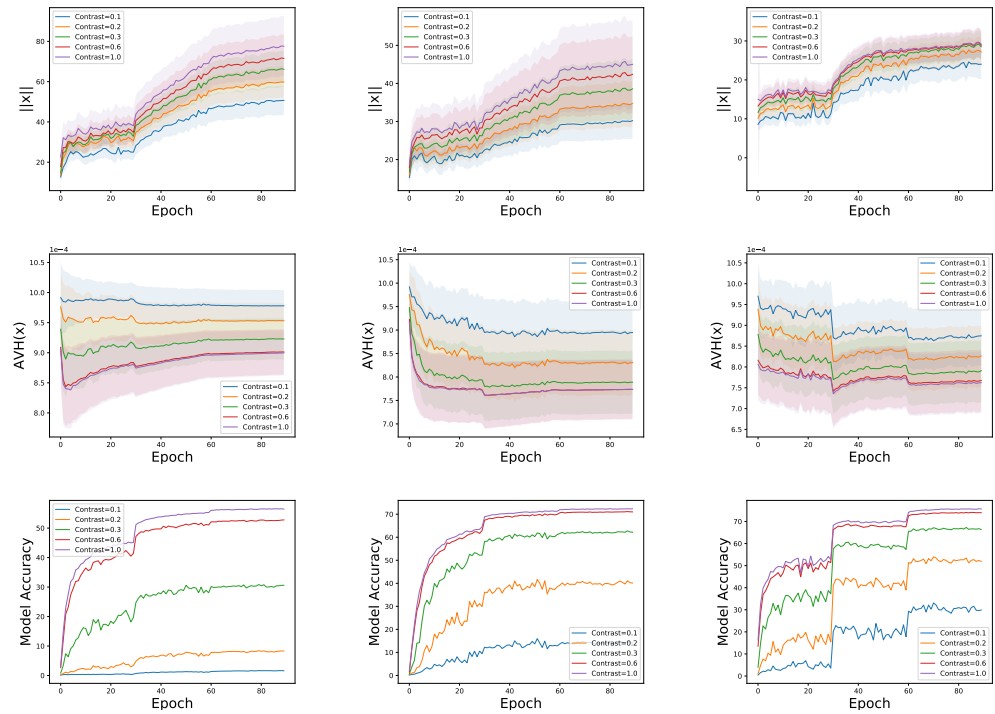

Figure 10: The top three plots show the number of Epochs v.s. Average $\ell_2$ norm across ImageNet validation samples which are split into five bins based on image contrast degradation level information. The middle three plots represent number of Epochs v.s. Average AVH(x). The bottom ones present number of Epochs v.s. Model Accuracy. From left to right, we use AlexNet, Vgg19 and ResNet50.

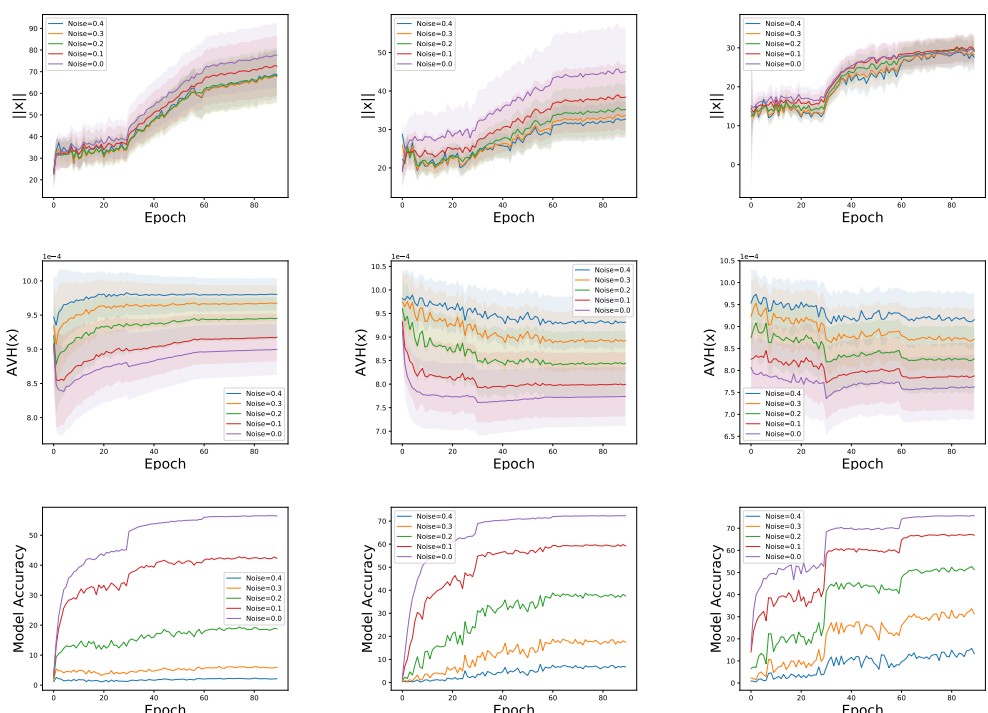

Figure 11: The top three plots show the number of Epochs v.s. Average $\ell_2$ norm across ImageNet validation samples which are split into five bins based on image noise degradation level information. The middle three plots represent number of Epochs v.s. Average AVH(x). The bottom ones present number of Epochs v.s. Model Accuracy. From left to right, we use AlexNet, Vgg19 and ResNet50.

## A.3    ADDITIONAL EXPERIMENTS FOR OBSERVING DYNAMICS ON MNIST

Figure 12 illustrates how the average norm of the feature embedding and angles between feature and class embedding for testing samples vary in 60 iterations during the training process. The average norm increases with a large initial slope but it flattens slightly after 10 iterations. On the other hand, the average angle decreases sharply at the beginning and then becomes almost flat after 10 iterations.

Moreover, we explore the difference between norm and angle change for easy and hard human examples in more details. Figure 13 also plots the angle and norm changes for two examples, which are hard and easy for human visualization, in the training phase. Note that both examples are testing data and thereby have never fit into the model. We can see that for the angle, both of them drop largely initially and then the angle for the easy one converges to a much lower value. For the norm, both of them are increasing drastically at an early stage but that for the harder example keeps climbing even when that for the easy one saturates.

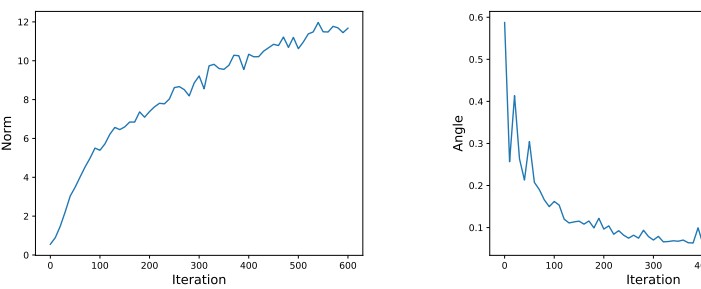

Figure 12: Average $\ell_2$ norm and angle of the embedding across all testing samples v.s. iteration number.

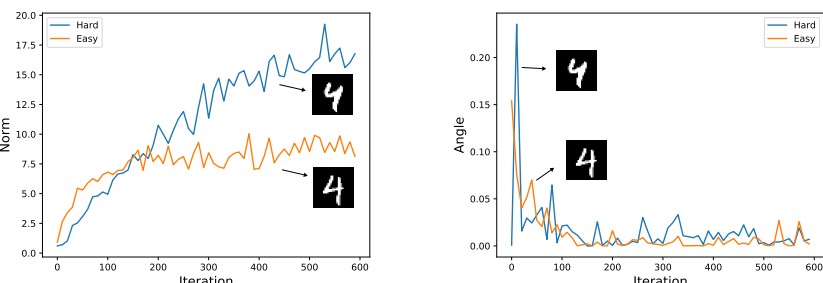

Figure 13: $\ell_2$ norm and angle of the embedding of an easy sample and a hard sample v.s. iteration number.

### A.4 ADDITIONAL EXPERIMENTS FOR TRAINING DYNAMICS ON CIFAR10 AND CIFAR100

Figure 14 and 15 show the dynamics of average $\ell_2$ norm of the embeddings and average AVH(x) on CIFAR10 and CIFAR100 datasets respectively. We can observe the similar phenomenons we have discussed in section 4. It further supports our theoretical foundation from [61] that gradient descent converges to the same direction as maximum margin solutions irrelevant to the $\ell_2$ norm of classifier weights or feature embeddings.

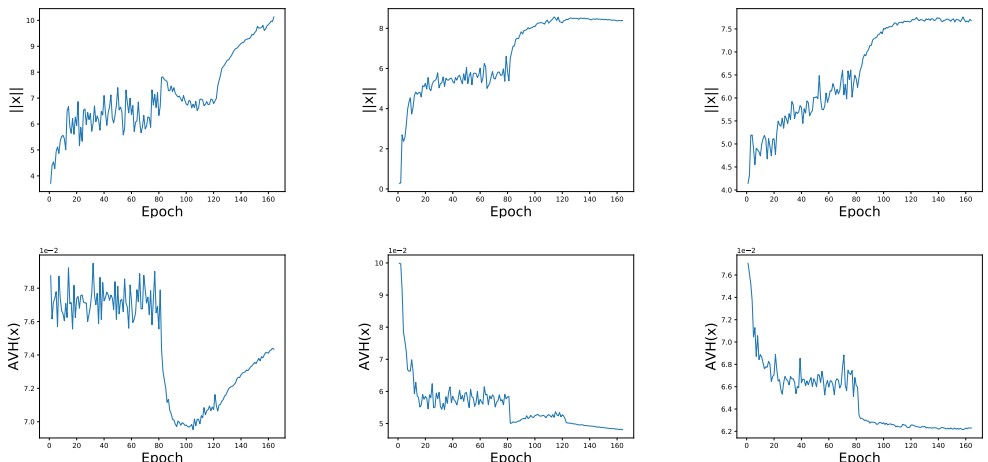

Figure 14: The top three plots show the number of Epochs v.s. Average $\ell_2$ norm across CIFAR10 validation samples. The bottom three plots represent number of Epochs v.s. Average AVH(x). From left to right, we use AlexNet, Vgg19 and ResNet50.

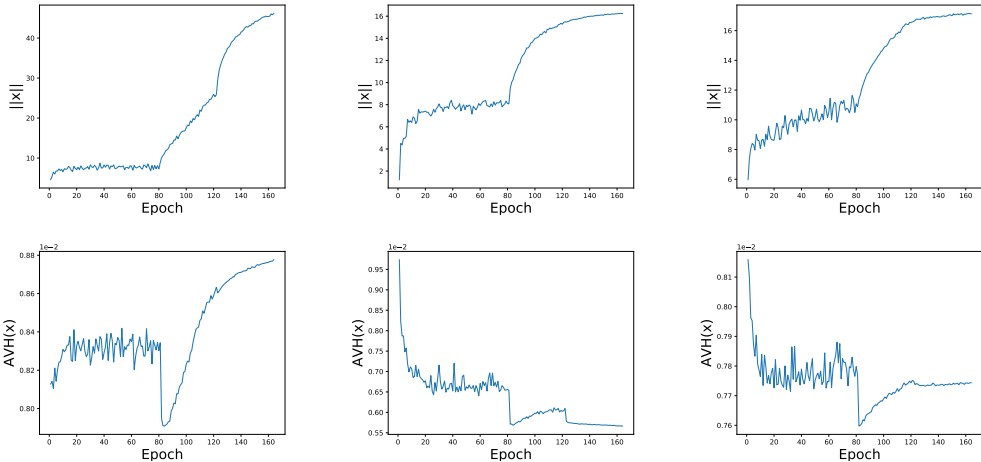

Figure 15: The top three plots show the number of Epochs v.s. Average $\ell_2$ norm across CIFAR100 validation samples. The bottom three plots represent number of Epochs v.s. Average AVH(x). From left to right, we use AlexNet, Vgg19 and ResNet50.

### A.5 Additional Experiments for Training Dynamics on ImageNet

Figure 17 presents the dynamics of the average $\|x\|_2$ and the dynamics of the accuracy for validation samples vary in 90 epochs during the training on AlexNet, VGG19, DenseNet121 and ResNet50. In figure 16 and 18, we average over validation samples in five human selection frequency bins separately, and then compute the average embedding norm, AVH and model accuracies. In figure 11 and 19, we average over validation samples in five image noise degradation level bins separately, and then compute the average embedding norm, AVH and model accuracies. In figure 10 and 20, we average over validation samples in five image contrast degradation level bins separately, and then compute the average embedding norm, AVH and model accuracies. Figure 21 shows the training dynamics of the model confidence on AlexNet, VGG19 and ResNet50.

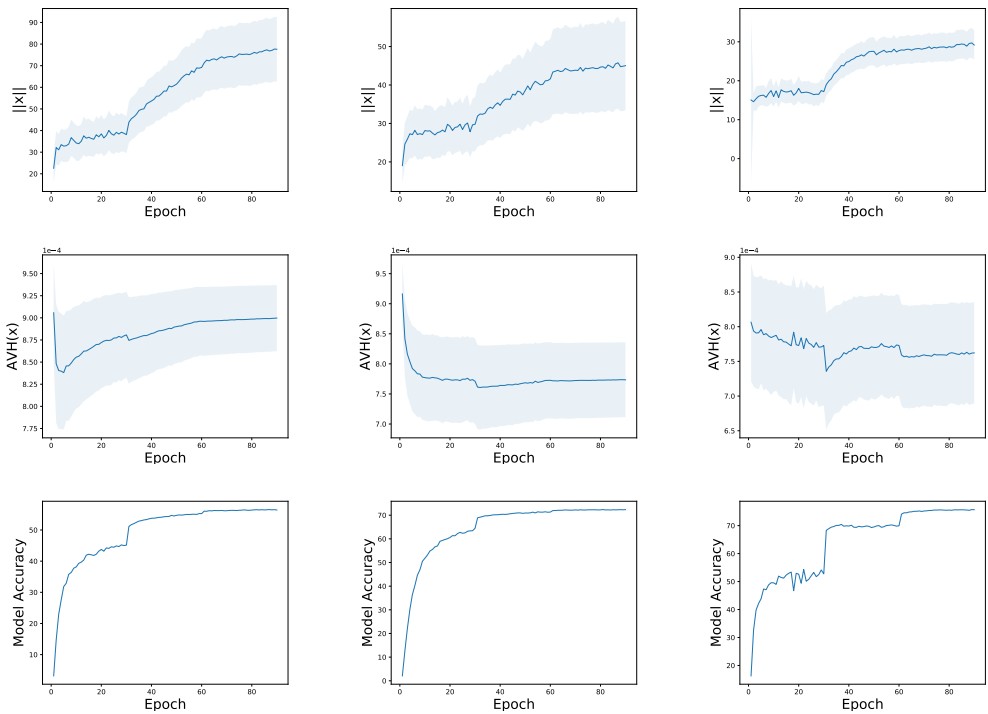

Figure 16: The top three plots show the number of Epochs v.s. Average $\ell_2$ norm across all ImageNet validation samples. The middle three plots represent number of Epochs v.s. Average AVH(x). The bottom ones present number of Epochs v.s. Model Accuracy. From left to right, we use AlexNet, Vgg19 and ResNet50. The plots for DenseNet are in Appendix.

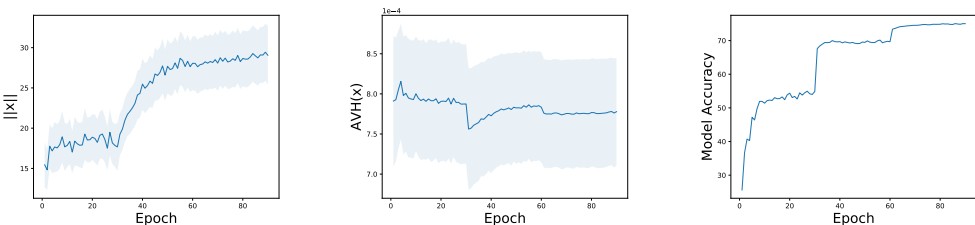

Figure 17: Average $\ell_2$ norm and angle of the embeddings of validation set v.s. number of epochs on DenseNet121.

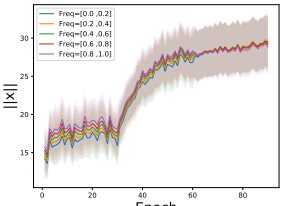 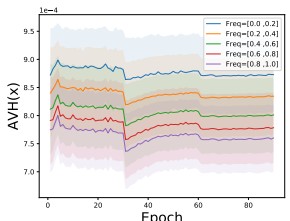 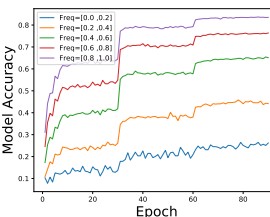

Figure 18: The left figure shows the number of epochs v.s. Average $\ell_2$ norm across ImageNet validation set which are split into five bins based on human selection frequency information. The middle one represents number of epochs v.s. average AVH(x). The right one present number of epochs v.s. model accuracy on DenseNet121.

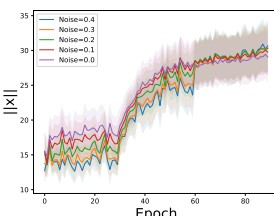 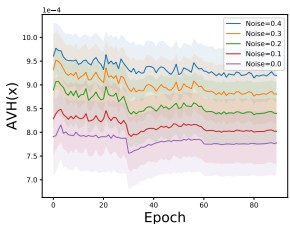 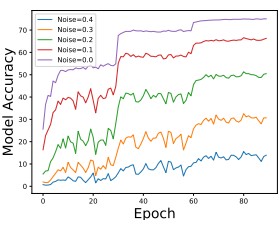

Figure 19: The left plot shows the number of Epochs v.s. Average $\ell_2$ norm across ImageNet validation set which is split into five bins based on image noise degradation level information. The middle plot represent number of epochs v.s. average AVH(x). The right one presents number of epochs v.s. model accuracy on DenseNet121.

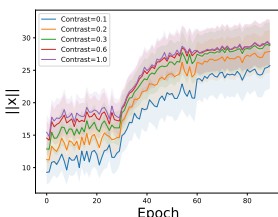 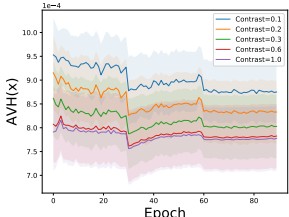 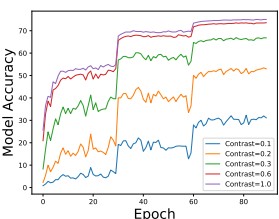

Figure 20: The left figure shows the number of epoch v.s. average $\ell_2$ norm across ImageNet validation set which is split into five bins based on image contrast degradation level information. The middle one represents number of epochs v.s. average AVH(x). The right one presents number of epochs v.s. model accuracy on DenseNet121.

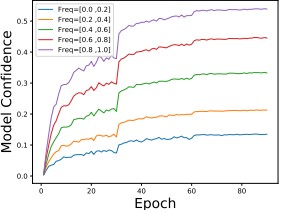 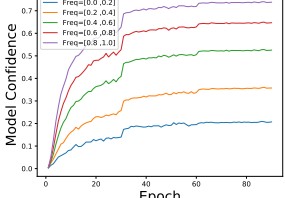 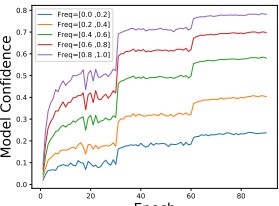

Figure 21: presents number of Epochs v.s. Model Confidence. From left to right, we use AlexNet, Vgg19 and ResNet50.

# B  A SPECIAL CASE: ADVERSARIAL EXAMPLES

We show a special case in Figure 22 to illustrate how the norm and the angle change when one sample switches from one class to another. Specifically, we change the sample from one class to another using adversarial perturbation. It is essentially performing gradient ascent to the ground truth class. In Figure 22, the purple line denotes the trajectory of an adversarial sample switching from one class to another. We can see that the sample will first shrink its norm towards origin and then push its angle away from the ground truth class. Such a trajectory indicates that the adversarial sample will first approach to the origin in order to become a hard sample for this class. Then the sample will change the angle in order to switch its label. This special example fully justifies the importance of both norm and angle in terms of the hardness of samples.

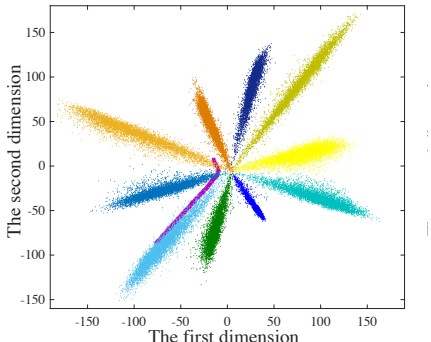

Figure 22: Trajectory of an adversarial example switching from one class to another. The purple line denotes the trajectory of the adversarial example.

## C    MORE DISCUSSIONS ON THE DIFFERENCE BETWEEN AVH AND MODEL CONFIDENCE

The difference between AVH and model confidence lies in the feature norm and its role during training. To illustrate the difference, we consider a simple binary classification case where the softmax score (i.e., model confidence) for class 1 is

$$\frac{\exp(\boldsymbol{w}_1\boldsymbol{x})}{\sum_i \exp(\boldsymbol{w}_i\boldsymbol{x})} = \frac{\exp(\|\boldsymbol{w}_1\|\|\boldsymbol{x}\|\cos(\theta_{\boldsymbol{w}_1,\boldsymbol{x}}))}{\sum_i \exp(\|\boldsymbol{w}_i\|\|\boldsymbol{x}\|\cos(\theta_{\boldsymbol{w}_i,\boldsymbol{x}}))}$$

where $\boldsymbol{w}_i$ is the classifier weights of class $i$, $\boldsymbol{x}$ is the input deep feature and $\theta_{\boldsymbol{w}_i,\boldsymbol{x}}$ is the angle between $\boldsymbol{w}_i$ and $\boldsymbol{x}$. To simplify, we assume the norm of $\boldsymbol{w}_1$ and $\boldsymbol{w}_2$ are the same, and then the classification result is based on the angle now. Once $\theta_{\boldsymbol{w}_1,\boldsymbol{x}}$ is smaller than $\theta_{\boldsymbol{w}_2,\boldsymbol{x}}$, the network will classify the sample $\boldsymbol{x}$ as class 1. However, in order to further minimize the cross-entropy loss after making $\theta_{\boldsymbol{w}_1,\boldsymbol{x}}$ smaller than $\theta_{\boldsymbol{w}_2,\boldsymbol{x}}$, the network has a trivial solution: increasing the feature norm $\|\boldsymbol{x}\|$ instead of further minimizing the $\theta_{\boldsymbol{w}_1,\boldsymbol{x}}$. It is obviously a much more difficult task to minimize $\theta_{\boldsymbol{w}_1,\boldsymbol{x}}$ rather than increasing $\|\boldsymbol{x}\|$. Therefore, the network will tend to increase the feature norm $\|\boldsymbol{x}\|$ to minimize the cross-entropy loss, which is equivalent to maximizing the model confidence in class 1. In fact, this also matches our empirical observation that the feature norm keeps increasing during training. Most importantly, one can notice that AVH will stay unchanged no matter how large the feature norm $\|\boldsymbol{x}\|$ is. Moreover, this also matches our empirical result that AVH easily gets saturated while model confidence can keep improving. Therefore, AVH is able to better characterize the visual hardness, since it is trivial for the network to increase feature norm. This is the fundamental difference between model confidence and AVH.

To get a more intuitive sense of how feature norm can affect the model confidence, we plot the value of the model confidence for two scenarios: $\theta_{\boldsymbol{w}_1,\boldsymbol{x}} < \theta_{\boldsymbol{w}_2,\boldsymbol{x}}$ and $\theta_{\boldsymbol{w}_1,\boldsymbol{x}} > \theta_{\boldsymbol{w}_2,\boldsymbol{x}}$. Under the case that the sample $\boldsymbol{x}$ belongs to class 1, once we have $\theta_{\boldsymbol{w}_1,\boldsymbol{x}} < \theta_{\boldsymbol{w}_2,\boldsymbol{x}}$, then we only need to increase the feature norm and can easily get nearly perfect confidence on this sample. In contrast, AVH will stay unchanged during the entire process and therefore is a more robust indicator for visual hardness than model confidence.

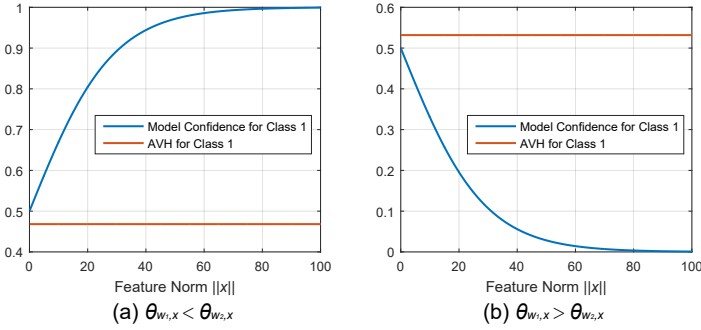

Figure 23: The comparison between AVH and model confidence when the feature norm keeps increasing. The figure is plotted according to the binary classification example discussed above. We assume $\|\boldsymbol{w}_1\| = \|\boldsymbol{w}_1\|$. When $\theta_{\boldsymbol{w}_1,\boldsymbol{x}} < \theta_{\boldsymbol{w}_2,\boldsymbol{x}}$, we use $\theta_1 = \pi/4 - 0.05$ and $\theta_2 = \pi/4 + 0.05$. When $\theta_{\boldsymbol{w}_1,\boldsymbol{x}} > \theta_{\boldsymbol{w}_2,\boldsymbol{x}}$, we use $\theta_1 = \pi/4 + 0.05$ and $\theta_2 = \pi/4 - 0.05$. Note that, unlike model confidence, the smaller AVH is, the more confident the network is (i.e., the easier the sample is).

