# OpenReview forum: "Angular Visual Hardness"
_ICLR.cc/2020/Conference — Reject_

### Official Review · AnonReviewer1 · 2019-10-24
**Official Blind Review #1**

**Rating:** 1

**Review:**

The paper proposes, when given a CNN, an image and its label, a measure called angular visual hardness (AVH). The paper shows that AVH correlates with human selection frequency (HSF) [RRSS19].

Pros:
1. The authors experimented with a representative set of trained models in my opinion. (More on this in Con2)
2. In Section 6, the authors acknowledge a substantive counter-example/argument. (More on this in Con2)

Cons (I put the ones that weighed the most on my decision first):
1. The presentation is confusing, and at times self-contradictory. For example, in Section 1, the paper asserts that “the two systems differ in what they view as hard examples that appear ambiguous or uncertain“ but then proceeds to claim that AVH being a CNN-derived quantity (more on this in Con2) correlates well with HSF. In fact, [RRSS19] (heavily cited here) seems to suggest exactly the opposite that harder examples for humans are also hard for CNNs. This is not very surprising as high accuracies of these CNNs imply their agreement with human judgment: we are learning a dataset labeled by humans. (More on this in Que2)
2. AVH is a function of a particular CNN (architecture and parameter values) and the target class label _in addition_ to the input image. These dependencies make AVH a measure of the ambiguity of an image very problematic. Granted that the paper presents evidence that AVH correlates with HSF for a number of _trained_ models but they will be of different values.
3. The work is not well self-contained. HSF, the core quantity studied is not introduced with sufficient details. (See Que1 and Sug1)

Some possible mistakes/typos:
1. Feature vector x and class weight w in general do not lie on S^n. Indeed your definition of A(u, v) only relies on u, v being nonzero.
2. There is a missing { above Definition 1.
3. In Definition 1, “for any x” -> for any (x, y).
4. In References, [10] is a duplicate of [11].
5. The captions in Figures 5, 6, and 7 in Appendix A might be wrong. They say ||x|| whereas the y-axis in the plots is labeled AVH.

Questions (I listed the important ones first):
1. What is human visual hardness (HVH)? How is HSF related to HVH? Why is being selected from a pool of images (in the procedure described in [RRSS19]) a good measure of HVH?
2. Since the class logit is exactly <x, w_c>, with arccos being a decreasing function, I expect AVH to behave very much like the opposite of model confidence (Definition 2). And this seems to be confirmed in Table 1 (performing a confidence calibration on validation set might increase this further). I wonder how AVH is different from model confidence _qualitatively_ and consequentially what insights do we gain (or should we expect to gain) by studying AVH instead of model confidence?
3. Degradation levels (DL) are mentioned early on but the experiments and figures were not shown in the main text (deferred till Appendix). What is the rationale?
4. The middle row in Figure 3 has a small range of ~1e-4. Is that expected? Can you provide some simple arguments? The closeness of the initial and final values of AVH in the AlexNet plot also concerns me.
5. How is the visualization in Figure 1 generated? It is not immediately clear to me how the high dimensional space is projected into 2D. My concern is that though suggestive in the figure, the category weights w_c in general do not spread out evenly. Do they? I would suggest reporting the angular separation of the category weights (maybe by showing them in a CxC matrix).
6. In Figure 2, what happens to the dark stripes? Are there no data points with the specified range of HSF values?

Minor issues (factored little to none in my decision):
1. There are 60+ citations but their relevance to the current seems questionable in many cases. Many of them are accompanied by little or no technical comparison when they are mentioned. In particular, in Section 2 on the related work from psychology/neuroscience, little specifics are discussed to contextualize the current work.
2. Many arguments come across as (highly) speculative and imprecise. As a result, I find the reasoning and logical story diluted and hard to follow.
3. The comparison with feature norm seems poorly motivated. The other quantities, namely AVH and model confidence, both depend on the class label.
4. The term hardness has a rich history and connotation in the algorithmic analysis literature. I would suggest using a different term, as the hardness of a problem usually reflects some intrinsic aspects of its structure and not dependent on some algorithm.

Suggestions:
1. If DL is not important to the core results, it will help simplify and focus the presentation by leaving them out entirely.
2. Try to be more concise and more precise in the presentation. It might also benefit from more formalism wherever possible, and more procedural details, when human studies or notions is involved. The latter seems to be a lesson from [RRSS19] (in regard to reproducibility).

In summary, I do not recommend accepting the current article.

(To authors and other reviewers) Please do not hesitate to directly point out my misunderstandings. I am open to acknowledging mistakes and revising my assessment accordingly.

**Experience Assessment:**

I have read many papers in this area.

**Review Assessment: Checking Correctness Of Derivations And Theory:**

N/A

**Review Assessment: Checking Correctness Of Experiments:**

I assessed the sensibility of the experiments.

**Review Assessment: Thoroughness In Paper Reading:**

I read the paper thoroughly.

---

> ### Author Response · Authors · 2019-11-12
> **Clarification to some misunderstandings**
>
>
> We sincerely thank the reviewer for the great efforts in helping us to improve clarity. We appreciate the constructive feedback as it helps the paper to better impact a wider range of audiences.
>
> However, there is a major misunderstanding from the reviewer which we would like to clarify first. The essence of such misunderstanding lies in confusing “dataset-level accuracy” with “sample-level model confidence”. Let us first provide a simple example here: There are only two labels 1 and 0. For image A, 3 out of 5 humans (HSF 0.6) vote for label 1; for image B, 5 out of 5 humans (HSF 1.0) vote for label 1. CNN only sees [A, 1] and [B, 1]. There is no information about how humans think the images are easy (HSF 1.0) or hard (HSF 0.6) passed to CNN. CNNs learn a dataset labeled by humans but the explicit hardness/confidence information is never revealed to them. Despite this AVH, which is coming purely from CNN correlated very well with HSF. While other popular measures like the softmax do not correlate well.  So the fact that such a measure exists and we can compute it is interesting in itself.
>
> We are not entirely sure what the reviewer means by the accuracy of CNN and hardness/confidence of the classifier are the same thing. Accuracy is an average over the whole data, in which confidence/hardness is only defined as instance level. So there is no accuracy for a particular instance (it is right or wrong) and hardness is only defined for the instance. These are unrelated terms. For example, is that one model can have very high overall accuracy (say 0.99) on a dataset but the confidence score (softmax) on one example can be 0.6.  We hope that this point is fairly clear.
>
> Before we delve into answering the reviewer's questions. We will first explain how AVH can be vastly different with model confidence for the reviewer to better understand AVH.
>
> **Difference between AVH and Model Confidence**
>
> The difference between AVH and model confidence lies in the feature norm and its role during training. To illustrate the difference, we consider a simple binary classification case where the softmax score (i.e., model confidence) for class 1 is $\frac{\exp(W_1X)}{\sum_i\exp(W_iX)}=\frac{\exp(\|W_1\|\|X\|\cos(\theta_{W_1,X}))}{\sum_i\exp(\|W_i\|\|X\|\cos(\theta_{W_i,X}))}$ where $W_i$ is the classifier weights of class $i$, $X$ is the input deep feature and $\theta_{W_i,X}$ is the angle between $W_i$ and $X$. To simplify, we assume the norm of $W_1$ and $W_2$ are the same, and then the classification result is based on the angle now. Once $\theta_{W_1,X}$ is smaller than $\theta_{W_2,X}$, the network will classify the sample $X$ as class 1. However, in order to further minimize the cross-entropy loss after making $\theta_{W_1,X}$ smaller than $\theta_{W_2,X}$, the network has a trivial solution: increasing the feature norm $\|X\|$ instead of further minimizing the $\theta_{W_1,X}$. It is obviously a much more difficult task to minimize $\theta_{W_1,X}$ rather than increasing $\|X\|$. Therefore, the network will tend to increase the feature norm $\|X\|$ to minimize the cross-entropy loss, which is equivalent to maximizing the model confidence in class 1. In fact, this also matches our empirical observation that the feature norm keeps increasing during training. Most importantly, one can notice that AVH will stay unchanged no matter how large the feature norm $\|X\|$ is. Moreover, this also matches our empirical result that AVH easily gets saturated while model confidence can keep improving. Therefore, AVH is able to better characterize the visual hardness and is also a more robust indicator to visual hardness than model confidence, since it is trivial for the network to increase the feature norm. This is a fundamental difference between model confidence and AVH.
>
> We update an additional *Appendix C* to better illustrate the difference between AVH and model confidence. We also plot the quantitative difference between model confidence and AVH.

---

> > ### Author Response · Authors · 2019-11-12
> > **Responses to individual questions (Part 1)**
> >
> >
> > **Question 1a): “The presentation is confusing, and at times self-contradictory. For example, in Section 1, the paper asserts that “the two systems differ in what they view as hard examples that appear ambiguous or uncertain“ but then proceeds to claim that AVH being a CNN-derived quantity (more on this in Con2) correlates well with HSF.”
> >
> > **Response:  Thanks for pointing out the issue. We have revised our introduction to improve clarity. Here we specifically mean that the softmax probability output, which is a popular confidence measure adopted by CNNs, does not match well with HSF. But this does not contradict the conclusion that CNNs can derive other quantities that correlates well with HSF.
> >
> > **Question 1b): “In fact, [RRSS19] (heavily cited here) seems to suggest exactly the opposite that harder examples for humans are also hard for CNNs. This is not very surprising as high accuracies of these CNNs imply their agreement with human judgment: we are learning a dataset labeled by humans.”
> >
> > **Response: Thanks for raising the concern. As mentioned in the beginning, this paper focuses on the better measurement of sample-level confidence instead of dataset-level accuracies. In light of this, the reviewer’s conclusion that “high accuracies of these CNNs imply their agreement with human judgment: we are learning a dataset labeled by humans.” seems irrelevant to this work. In addition, from their experiments (section 4 in [RRSS19]) as well as the direct confirmation from the authors, by saying “images selected less frequently by the MTurk workers are harder for the models”, they are referring to correlation with dataset-level accuracy (CNNs do make mistakes on images ambiguous to human) instead of sample-level confidence (CNNs being over-confident on wrong samples). Thus their conclusion is not contradicting the technical correctness of this work.
> >
> > **Question 2): “AVH is a function of a particular CNN (architecture and parameter values) and the target class label _in addition_ to the input image. These dependencies make AVH a measure of the ambiguity of an image very problematic. Granted that the paper presents evidence that AVH correlates with HSF for a number of _trained_ models but they will be of different values. ”
> >
> > **Response:  AVH is not defined only for some particular architectures of CNN. As long as the architecture has embedding layer and classifier, AVH can be computed. In addition, as the reviewer mentioned, we use four different CNN architectures for all the experiments to show the consistency of our observations.
> >
> > If we understand the reviewer’s confusion correctly, it means “for one particular picture, different models have different AVH scores is a problem”.  This is actually an advantage of AVH instead of a problem. AVH scores are not the same since different models have different generalization abilities and naturally tend to show different levels of confidence. We provide more details in Observation 4 on Page 7.
> >
> > **Question 3): “The work is not well self-contained. HSF, the core quantity studied is not introduced with sufficient details.”
> >
> > **Response: We have a formal definition of HSF in “Page 3 Definition 3”.
> >
> > **Action Taken: We add the definition of HVH and detailed discussion about how we relate HVH and HSF in the second paragraph of our Introduction.

---

> > > ### Author Response · Authors · 2019-11-12
> > > **Responses to individual questions (Part 2)**
> > >
> > >
> > > **Question 4): “What is human visual hardness (HVH)? How is HSF related to HVH? Why is being selected from a pool of images (in the procedure described in [RRSS19]) a good measure of HVH?”
> > >
> > > **Response: Human visual hardness is a measure of how hard it is for humans to classify a particular picture correctly. Human selection frequency is a proxy for this measure. It is a reasonably good proxy because humans tend to select the pictures which can be easily identified as a given label and might miss the ones which are hard for them to identify from a pool of pictures.
> > >
> > > **Action Taken: We add the definition of HVH and detailed discussion about how we relate HVH and HSF in the second paragraph of our Introduction.
> > >
> > >
> > > **Question 5): “Since the class logit is exactly <x, w_c>, with arccos being a decreasing function, I expect AVH to behave very much like the opposite of model confidence (Definition 2). And this seems to be confirmed in Table 1 (performing a confidence calibration on validation set might increase this further). I wonder how AVH is different from model confidence _qualitatively_ and consequently what insights do we gain (or should we expect to gain) by studying AVH instead of model confidence?”
> > >
> > > **Response: First of all, Table 1 shows that AVH correlates (significantly) more closely to human selection frequency than model confidence, supporting our claim that AVH can better reflect the human visual hardness. Also, we use Spearman's correlation coefficients which is a statistical measure of the strength of a monotonic relationship between paired data.
> > >
> > > There are several obvious insights we can gain from AVH:
> > >
> > > First, AVH demonstrates that angles serve as a more calibrated similarity measure that better reflects human perception. However, one can also see that angles are generally difficult to optimize in current neural networks, indicating that current loss functions or even network architectures are not well designed to optimize the angles. These observations open up a new research direction to improve our network architectures and objective functions towards better optimization of angular similarity instead of the inner product.
> > >
> > > Second, AVH reveals an interesting inductive bias in CNNs: features learned by CNNs tend to be discriminative on a hypersphere manifold. The discovery of such inductive bias could be useful for improving the accuracy and robustness of CNNs in all kinds of applications.
> > >
> > > Last, AVH produces a more calibrated and accurate visual hardness measure, facilitating the applications like self-training for domain adaptation (shown in Section 5 in our paper), robust learning against noisy labels, etc. We have already shown in our paper that AVH can significantly boost the state-of-the-art performance in domain adaptation by outperforming the previous state-of-the-art result (in ICCV 2019) by more than 3%.
> > >
> > >
> > > **Question 6): Degradation levels (DL) are mentioned early on but the experiments and figures were not shown in the main text (deferred till Appendix). What is the rationale?
> > >
> > > **Response: Because the DL results are consistent with HSF results and because of the space limit, we attach the results in the appendix.
> > >
> > > **Action Taken: We move everything about DL to the appendix and only link the similar results in the related work and section 3.1 notation and setup.

---

> > > > ### Author Response · Authors · 2019-11-12
> > > > **Responses to individual questions (Part 3)**
> > > >
> > > >
> > > > **Question 7): The middle row in Figure 3 has a small range of ~1e-4. Is that expected? Can you provide some simple arguments? The closeness of the initial and final values of AVH in the AlexNet plot also concerns me.
> > > >
> > > > **Response: Yes. AVH is basically an angle normalized by the sum of all angles. Because ImageNet has 1000 classes, it is reasonable that AVH is in the range of 1e-4-1e-3. For AlexNet, a) the model is comparatively a poor model and b) AVH does not necessarily need to decrease too much to have a high impact on the accuracy because as long as the angle between the embedding and the correct class the smallest among all other angles, it will produce a correct prediction (it can be very close to the decision boundary).
> > > >
> > > > **Question 8): How is the visualization in Figure 1 generated? It is not immediately clear to me how the high dimensional space is projected into 2D. My concern is that though suggestive in the figure, the category weights w_c, in general, do not spread out evenly. Do they? I would suggest reporting the angular separation of the category weights (maybe by showing them in a CxC matrix).
> > > >
> > > > **Response: Figure 1 does *NOT* use any projections or visualization tools (e.g., T-SNE, etc.) because the embedding dimension is originally 2-dimension. Specifically, we directly set the embedding layer (the layer before the classifiers) of CNN to 2 dimensions. That means we use 10 classifiers (each classifier is also 2-dimensional) to classifier 2-dimensional features learned by a CNN. Therefore the category weights do spread out evenly.
> > > >
> > > > **Action Taken: We revise the caption for Figure 1 to make it more clear. We will also upload the code to reproduce the visualization.
> > > >
> > > > **Question 9): In Figure 2, what happens to the dark stripes? Are there no data points with the specified range of HSF values?
> > > >
> > > > **Response: Yes there are no data points for the values within those ranges.
> > > >
> > > > We hope the above response can help the reviewer better understand our paper. We sincerely thank the reviewer again for all the questions and suggestions, which greatly improve the clarity of our work.

---

### Official Review · AnonReviewer2 · 2019-10-24
**Official Blind Review #2**

**Rating:** 8

**Review:**

This paper defined Angular Visual Hardness (AVH) based on the angle between image feature embedding and the weights of the target class. The authors compared the correlation between AVH and human selection frequency with model confidence and feature norm. The results show that both AVH and model confidence have correlation, but AVH has a stronger correlation than model confidence. Differently from the conjecture of [41], feature norm was not correlated with human selection frequency.

Next, the training dynamics of AVH are analyzed. The results show that feature norm keeps increasing during training, whereas AVH hits a plateau very early even when the accuracy or loss is still improving. Also, AVH correlates human selection frequency across different deep models, and it also correlates the model’s accuracies.

As an application of AVH, the authors applied it to sample selection of self-training for domain adaption. The proposed selection method based on AVC could improve a state-of-the-art self-training method, of which sample selection is based on model confidence of CNN.

Overall, the experimental contribution of this paper is good, and the experimental conditions seem to be correct.

Minor problems.
In the analysis of the dynamics of training, the authors compared the AVH with feature norm. How about the dynamics of model confidence? Is it similar to the feature norm?

The curves of different levels of hardness are missing in Fig.14.



**Experience Assessment:**

I have read many papers in this area.

**Review Assessment: Checking Correctness Of Derivations And Theory:**

N/A

**Review Assessment: Checking Correctness Of Experiments:**

I assessed the sensibility of the experiments.

**Review Assessment: Thoroughness In Paper Reading:**

I read the paper at least twice and used my best judgement in assessing the paper.

---

> ### Author Response · Authors · 2019-11-13
> **Response to Reviewer2**
>
>
> Thanks for your recognition on our work. With our extensive empirical study, AVH is shown to be able to better identify the visual hardness than model confidence. AVH is generally useful in a number of applications, not limited to self-training for domain adaptation. Potential applications include robust learning against noisy labels, semi-supervised learning, etc.
>
> **Question: In the analysis of the dynamics of training, the authors compared the AVH with feature norm. How about the dynamics of model confidence? Is it similar to the feature norm?
>
> **Response: The dynamics of the model confidence during training is similar to model accuracy, and its curve is indeed close to the curve of feature norm. We have put the figures of the model confidence (on AlexNet, VGG19, and ResNet50) in Appendix A.3.
>
> The difference between AVH and model confidence lies in the feature norm and its role during training. To illustrate the difference, we consider a simple binary classification case where the softmax score (i.e., model confidence) for class 1 is $\frac{\exp(W_1X)}{\sum_i\exp(W_iX)}=\frac{\exp(\|W_1\|\|X\|\cos(\theta_{W_1,X}))}{\sum_i\exp(\|W_i\|\|X\|\cos(\theta_{W_i,X}))}$ where $W_i$ is the classifier weights of class $i$, $X$ is the input deep feature and $\theta_{W_i,X}$ is the angle between $W_i$ and $X$. To simplify, we assume the norm of $W_1$ and $W_2$ are the same, and then the classification result is based on the angle now. Once $\theta_{W_1,X}$ is smaller than $\theta_{W_2,X}$, the network will classify the sample $X$ as class 1. However, in order to further minimize the cross-entropy loss after making $\theta_{W_1,X}$ smaller than $\theta_{W_2,X}$, the network has a trivial solution: increasing the feature norm $\|X\|$ instead of further minimizing the $\theta_{W_1,X}$. It is obviously a much more difficult task to minimize $\theta_{W_1,X}$ rather than increasing $\|X\|$. Therefore, the network will tend to increase the feature norm $\|X\|$ to minimize the cross-entropy loss, which is equivalent to maximizing the model confidence in class 1. In fact, this also matches our empirical observation that the feature norm keeps increasing during training. Most importantly, one can notice that AVH will stay unchanged no matter how large the feature norm $\|X\|$ is. Moreover, this also matches our empirical result that AVH easily gets saturated while model confidence can keep improving.  Therefore, AVH is able to better characterize the visual hardness and is also a more robust indicator to visual hardness than model confidence, since it is trivial for the network to increase the feature norm.
>
> **Action Taken: We have put the figures of the model confidence (on AlexNet, VGG19, and ResNet50) in the Appendix A.3. We also update an additional Appendix C to further illustrate the difference between AVH and model confidence.
>
> **Question: The curves of different levels of hardness are missing in Fig.14.
>
> **Action Taken: The caption in Figure 14 is confusing. It is the average over all the samples not different hardness levels. Thanks for pointing it out and we have corrected it.

---

### Official Review · AnonReviewer3 · 2019-11-04
**Official Blind Review #3**

**Rating:** 8

**Review:**

Main Contribution:

This paper is trying to bridge the gap between CNN and the human  visual system by proposing a metric  (angular visual distance) and validate that this metric is correlated to the human visual hardness and this metric has a stronger relation  compared to the softmax score which has been viewed as a metric measuring the hardness of images in CNNs. Furthermore, this paper proposed a reasonable explanation for this observation, i.e., the norm is possibly not correlated to the human visual hardness and validate through the experiment. Finally, this paper shows that this metric is also useful  in other applications.

Innovative Part:

The metric proposed in this paper is based on an interesting and also innovative observation that  samples in each class will concentrate in a convex cone in the embedding space (e.g., shown in Figure 1) and the norm has no information on the visual hardness. I like this observation since several existing theoretical results have similar implications although in far simpler settings. For example, [1] shows that for LINEAR model with logistic loss, gradient descent converges to the maximum margin classifier while the norm (corresponding to ||x||_2 in this paper) diverges to infinity with log(T) rate.  If we are looking into the Figure 1, we will see that ten convex cones almost form an equal partition of  the two-dimensional space and this indicates that the classifier is very similar to the classifier with the maximum margin in the angular space (NOT in the Euclidean space). The observation is quite intuitive and has strong theoretical foundation, which is the main reason that I vote for the acceptance of this paper.

Drawbacks:

This paper also have several drawbacks but I do believe they can be addressed very easily.

1. The introduction is not well-written, especially the second paragraph. I strongly recommend modifying the introduction.

For the first three sentences of the second paragraph, do you mean that CNNs are constructed based on some properties of the human visual systems and thus they should have had some connections but they actually fundamentally differ in practice? Otherwise, if these two are fundamentally different with each other, what is the point of showing some connections between them?

For the sentence "we use this dataset to verify our hypothesis", what is the hypothesis? Do you mean the hypothesis that human visual hardness should have had connections to the classifying hardness for CNNs?

 For the sentence "Given a CNN, we propose a novel score function that has strong correlation with human visual hardness", I am not sure whether the word "strong" can be used here.

2. In table 1, I am not sure whether the author should assume that all audiences have some background on z-score, although I can understand it. I would also encourage the authors to use other correlation metrics with more intuitive explanations (e.g., correlation coefficients).

3. For the experiment, I would like to recommend authors adding the following experiments.

3.1) Show that on other datasets (e.g., CIFAR 10, 100), AVH converges fast to a plateau while the norm constantly diverges to infinity.
3.2) Introducing several other measurements to show the correlation.
3.3) I also would like to see similar results in Table 1 for different models.






[1] Soudry, Daniel, et al. "The implicit bias of gradient descent on separable data." The Journal of Machine Learning Research 19.1 (2018): 2822-2878.

**Experience Assessment:**

I have published one or two papers in this area.

**Review Assessment: Checking Correctness Of Derivations And Theory:**

I assessed the sensibility of the derivations and theory.

**Review Assessment: Checking Correctness Of Experiments:**

I carefully checked the experiments.

**Review Assessment: Thoroughness In Paper Reading:**

I read the paper at least twice and used my best judgement in assessing the paper.

---

> ### Author Response · Authors · 2019-11-15
> **Response to Reviewer3**
>
> Thank you for your recognition of our work. We appreciate your constructive suggestions, especially mentioning the existing theoretical results which can be extremely useful for strengthening the motivation of our work. We have added that in our introduction.
>
> **Question: The introduction is not well-written, especially the second paragraph
>
> **Action Taken: We have revised our logic and writing of our introduction addressing all your problems. Thanks for pointing it out.
>
> **Question: In table 1, I am not sure whether the author should assume that all audiences have some background on z-score, although I can understand it. I would also encourage the authors to use other correlation metrics with more intuitive explanations (e.g., correlation coefficients).
>
> **Response: We do provide the correlation coefficient in the second column of Table 1.
>
> **Action Taken: We have made the column name more explicit and add more details about the testing under hypothesis 3.
>
> **Question: For the experiment, I would like to recommend authors adding the following experiments.
>
> **Action Taken:
> We have added the experiments of training dynamics on both CIFAR10 and CIFAR 100 in Appendix A.4 (Page 20).
> The experiments provide similar supportive results to our claims in the training dynamics section.
>
> We have added Pearson and Kendall Tau correlation coefficients.
>
> We have added four tables in Appendix A1 (Page 15-16)  for all three types of correlation coefficients for four different models (AlexNet, VGG19, ResNet50, and DenseNet121). The experiments also provide similar supportive results to our hypotheses. We found out that better models, like ResNet and DenseNet, correlates stronger with Human Selection Frequency.  Also, the difference between the model confidence correlation coefficient and AVH correlation coefficient with HSF is also larger for better models.  It helps verify our claim that AVH is an indicator of model generalizability.
>
> Thanks again for all the suggestions which help us improve the completeness of our paper.

---

### Author Response · Authors · 2019-11-15
**Summary of the Revision**

We sincerely thank all the reviewers for providing constructive suggestions and helping us improve the paper! We list the major changes we have done according to the recommendations in the following:

1. We made a revision to our introduction, especially the first and second paragraphs to better motivate our work and improve the logic flow. [Introduction]

2. We added a paragraph in the introduction to discuss Reviewer_3's suggested theoretical foundation. [Introduction]

3. We provide more details about the correlation testings. [Section 3 Hypothesis 3]

4. We have moved all the experiments and discussions about image degradation to the appendix. [Appendix A.2 & A.5]

4. We added a section to better illustrate the difference between AVH and model confidence. We also plot the quantitative difference between model confidence and AVH in that section. [Appendix C]

5. We added the figures of the training dynamics for model confidence, which help confirm the different between AVH and model confidence. [Appendix A.3(Figure 21)]

6. We added training dynamics experiments on CIFAR10 and CIFAR100 to confirm that our observation of the norm and angle is not a phenomenon that is only on ImageNet. [Appendix A.4 (Page 20)]

7. We added 2 other correlation coefficient testings, Pearson and Kendall Tau. We also added the testings on all four models. [Appendix A.1 (Page 15-16) ]

8. We fixed the typos and citation issues.

---

### Decision · Program_Chairs · 2019-12-19

**Decision:**

Reject

**Comment:**

This paper proposes a new measure for CNN and show its correlation to human visual hardness. The topic of this paper is interesting, and it sparked many interesting discussions among reviews. After reviewing each others’ comments, reviewers decided to recommend reject due to a few severe concerns that are yet to be address. In particular, reviewer 1 and 2 both raised concerns about potentially misleading and perhaps confusing statements around the correlation between HSF and accuracy. A concrete step was suggested by a reviewer - reporting correlation between accuracy and HSF. A few other points were raised around its conflict/agreement with prior work [RRSS19], or self-contradictory statements as pointed out by Reviewer 1 and 2 (see reviewer 2’s comment). We hope authors would use this helpful feedback to improve the paper for the future submission.